# Wheel Hub Defects Image Recognition Base on Zero-Shot Learning

Xiaohong Sun [1,2], Jinan Gu [1,*], Meimei Wang [2,*], Yanhua Meng [2] and Huichao Shi [2]

1  School of Mechanical Engineering, Jiangsu University, Zhenjiang 212000, China; 20160269@ayit.edu.cn
2  School of Mechanical Engineering, Anyang Institute of Technology, Anyang 455000, China; 20200040@ayit.edu.cn (Y.M.); 20160267@ayit.edu.cn (H.S.)
*  Correspondence: gujinan@tsinghua.org.cn (J.G.); 20160297@ayit.edu.cn (M.W.)

**Abstract:** In the wheel hub industry, the quality control of the product surface determines the subsequent processing, which can be realized through the hub defect image recognition based on deep learning. Although the existing methods based on deep learning have reached the level of human beings, they rely on large-scale training sets, however, these models are completely unable to cope with the situation without samples. Therefore, in this paper, a generalized zero-shot learning framework for hub defect image recognition was built. First, a reverse mapping strategy was adopted to reduce the hubness problem, then a domain adaptation measure was employed to alleviate the projection domain shift problem, and finally, a scaling calibration strategy was used to avoid the recognition preference of seen defects. The proposed model was validated using two data sets, VOC2007 and the self-built hub defect data set, and the results showed that the method performed better than the current popular methods.

**Keywords:** image recognition; zero-shot learning; projection domain shift; hubness problem

## 1. Instruction

The automobile is an indispensable means of transportation in people's daily life, it is also an important part of the national economy, and the wheel hub is one of the key parts of the automobile. With the development of China's automobile industry, the technical level of wheel hub manufacturing enterprises is constantly improving, and product export volume is large and continues to grow. However, due to the rapid growth of production and the imperfection of processing technology, more than 40 defects have been found in wheel hub products. These defects not only affect the good appearance and brand image of products but also lead to serious traffic accidents. Therefore, researchers have done some work, such as Li [1], in order to improve the accuracy of automobile wheel hub defect image detection and recognition, an improved peak algorithm—the trend peak algorithm—was proposed to extract the wheel hub defect area and combined with the BP neural network to recognize the wheel hub defects. Zhang [2] aimed at the internal defects such as air hole and shrinkage cavity in the process of low-pressure casting of the wheel hub, a method of defect extraction based on mathematical morphology was employed, all of which belongs to the traditional recognition method's artificial design feature, so in the face of complex samples, the robustness is poor. While deep learning technology has advantages: automatic feature extraction, weight sharing, and needless image preprocessing. For example, Han [3] used Faster R-CNN with ResNet-101 as the target detection algorithm to detect scratches and points on the wheel hub. Sun [4] put forward an improved Faster R-CNN recognition model for multiple types of wheel hub defects, that is, by improving the shared network ZFNet to two revised branches (RPN and Fast R-CNN), and then four typical wheel hub defects are identified. Although the methods based on deep learning have made some progress, we find that existing supervised learning strategies rely on a large amount of labeled data. However, some defect samples are scarce or none at all, so they cannot meet

the requirements of the deep network training, which leads to the performance breakdown. At present, there are some different solutions, such as transfer learning [5], self-taught learning [6], and few-shot learning [7]. However, these methods are unable to cope with the condition of zero samples. When the test samples are never seen during training, which is called zero-shot learning (ZSL) [8], as shown in Figure 1.

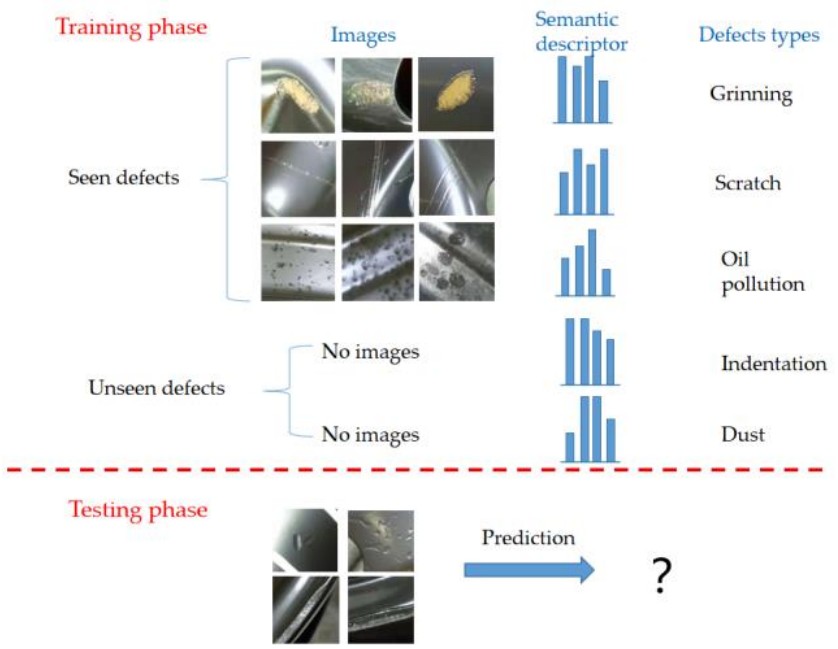

**Figure 1.** Zero-shot learning process. In training, there are seen classes with a large number of labeled images, but unseen classes with no labeled images, however, semantic descriptors for both seen and unseen classes are available, through which unseen classes are identified.

As we know, humans have the ability to retain knowledge from past learning tasks and use that knowledge to quickly integrate and solve new recognition tasks. Specifically, humans can easily identify these rare categories by using semantic description and their relationship to the seen classes. For example, a person could identify a new species "penguin" by the semantic description, "a flightless bird living in the Antarctic", or "kangaroo" by the portrayal, "an Australian animal with a pouch on its body", inspired by which we try to empower machines with the same intelligence, in the process of achieving this goal, zero-shot learning is essential to the realization of machine intelligence.

ZSL has broad prospects in the fields of autonomous driving [9], medical imaging analysis [10], intelligent manufacturing [11], robotics [12], etc. In these fields, although it is difficult to obtain new labeled images, advanced semantic descriptions of categories can be easily obtained.

In order to identify unseen classes, usually, a large scale of labeled samples (seen classes) is needed to train the deep model, and then the model is adapted to an unseen one. For ZSL, the seen and unseen classes are associated through a high-dimensional vector space of semantic descriptors. Each class corresponds to a unique semantic descriptor. Semantic descriptors can take the form of manually defined attributes [13] or automatically extracted word vectors [14]. Figure 1 shows the training and testing process of ZSL.

## 2. Related Work

There are three key technologies in zero-shot learning. First, the extraction method of image features: in the real world, image data is complex, redundant, and ever-changing, so the acquisition of image features plays an important role in narrowing the gap between image learning and high-level semantics. Second, the construction of semantic embedding space, different semantic embedding spaces usually reflect the different semantic properties

of object labels, therefore constructing a proper semantic embedding space and keeping its consistency with the image feature space is crucial to solving the zero-shot learning problem. Third, the association mode between image feature space and semantic embedding space, in the zero-shot learning method based on semantic embedding space, the association mode between both is the focus of most models.

### 2.1. Extraction Method of Image Features

In machine learning tasks, especially the classification problem, raw data need to be converted into a format that can be efficiently processing by computer. Traditional methods adopt artificial design features, but they require a large amount of manpower and rely on very specialized knowledge, therefore, they are not conducive to promotion in the whole field.

For visual applications, researchers have also proposed many distinctive feature learning methods, such as significant visual features based on the cognitive psychology model [15], efficient coding visual features for mobile computing platforms [16], etc. However, these methods generally have two limitations: first, they are usually unable to go beyond the feature expression of artificial design. Second, both automatic and manual design of image features belong to lower-level expression, lacking the ability to associate high-level semantic meaning, which leads to a deeper semantic gap between image features and high-level semantic attributes, that is why zero-shot learning performance was generally lower in 2008 to 2012.

In order to improve the semantic properties of image features, a deep convolutional neural network (CNN) [17] can be used to separate the complex and entangled factors hidden in the image through some simple and nonlinear models, then transforming the original image data into higher-level and more abstract feature expressions. These features have a strong expressive ability, which can not only reflect the semantic information of objects to a certain extent but also express the global features and context information of images, therefore features from CNN have a higher judgment ability among different objects. In view of the great success of deep feature in many research fields such as large-scale object recognition [18,19] and video recognition [20], deep feature has also been widely used in zero-shot learning [21,22] since 2012. Currently the VGG [23], GoogLeNet [23] model, ResNet [24] models are three deep feature extraction models.

### 2.2. Construction of Semantic Embedding Space

As a shared intermediate semantic layer, semantic embedding space bridges the semantic gap [25–27] between the underlying image feature space and the high-level semantic space, transcends the class boundaries between mutually exclusive objects, which is the key to solve the zero-shot learning problem. For zero-shot learning, semantic embedding spaces [28,29] are usually constructed independently of visual recognition tasks, that is an object class label, we represent an independent label as an interrelated label embedding vector by means of knowledge that is relatively easy to obtain in other fields.

Semantic vectors must meet two basic requirements: sufficient semanticity [30] and strong judgment ability [31], however, both are contradictory. Strong semanticity usually means that the semantic embedding space contains more details of the object, including information unrelated to the classification task, such as non-visual information, which leads to weak judgment. Strong judgment ability means that visual information that can best distinguish between different object classes are only focused on and ignore other information, therefore reducing semanticity. How to grasp a balance between the both is one of the concerns in the field of zero-shot learning.

Intuitively, we can manually define the attributes [32] of class labels according to expert knowledge, or automatically learn label semantic vectors [33] using machine learning technology. These two methods are the core routes widely used in the current zero-shot learning problem, and they have their own advantages and disadvantages, specifically, the artificial way can define the shared semantic expression with adjudicative and semantic

properties, so it performs very well in zero-shot learning. However, this way requires a lot of time and relies heavily on professional knowledge, so it is not easy to popularize for more learning problems. The automatic method is fast to construct a semantic embedding space, and theoretically, it can learn semantic vectors of large-scale classes, so it can be easily extended to large scale zero-shot learning. From the point of the whole field development situation, between 2008 and 2012, we mainly use the way of artificial building semantic embedding space, but since 2012, with the size of the object classification tasks getting bigger, the automated way has gradually prevailed. One of the reasons is that the rise of deep learning technology has promoted NLP performance significantly. From the perspective of development, considering the trend of the object classification tasks, the main strategy in the future is still the automatic way.

In the field of zero-shot learning, there are two ways to construct semantic embedded space: manually defined semantic attributes and automatically learned semantic word vectors. The former is the most commonly used and very effective way to construct semantic embedded space. It uses artificially defined, shared natural language attributes, which can be used to annotate object classes. In view of the shared nature of semantic attributes among object classes, we can learn attribute classifiers from the training data and then use this classifier to predict unseen classes. To be specific, for any unseen class image, we can first judge what attributes it has, and then compare it with the attribute annotation defined in advance, so as to determine that it belongs to the object class with the most similar attribute annotation.

The latter uses large, more readily available unsupervised text documents (store rich semantic information about object classes), and automatically constructs semantic embedding spaces for object classes based on machine learning techniques. Based on text documents, we usually convert object class tags to vector form. In NLP, the most intuitive expression form of a word vector is one-hot encoding, that is, the 0/1 vector containing only one 1 [0, 0, ..., 1, 0, ..., 0], where the vector length represents the size of the entire lexicon of the text documents.

The reason why early NLP tasks adopted one-hot encoding [34] is that it has a simple form, robustness, and a simple model based on a large amount of training data is superior to a complex model based on a small amount of training data. However, this approach has two disadvantages: (1) it is easy to cause dimensional disasters [35]. We know that large documents (billions of words, millions of words) are becoming more and more common in NLP tasks, which makes the length of word vectors in this form in millions, and it seriously troubles the development of the NLP algorithm, especially when it is applied to deep learning technology. (2) Semantic gap, which cannot describe the correlation between words. One-hot encoding considers that any two words are isolated from each other without semantic correlation.

### 2.3. Visual-Semantic Association Mapping Learning

Semantic embedding space is a bridge between image feature space and tag semantic space, and the establishment of vision-semantic association is an essential cornerstone of zero-shot learning, as shown in Figure 2.

In general, once the vision-semantic association is established, we can calculate the similarity between any unseen class and the unseen class prototype and classify the sample *x* based on the similarity. Among them, the quality of the vision-semantic relationship directly affects the performance of zero-shot learning. On the one hand, the vision-semantic relationship learned from the training data should have sufficient generalization ability, so that it can be applied to the unseen mutually exclusive data to the greatest extent. On the other hand, it should also have a strong judgment ability, so as to have a positive impact on the subsequent recognition process based on similarity.

We can model the vision-semantic relationship in the following three ways (as shown in Figure 2). The first one is to map image features to the semantic embedding space (we call it forward mapping, as shown in Figure 2a), and to identify unseen classes within the

semantic embedding space. In the second way, image features and class semantics are embedded and simultaneously mapped to a common space (known as a common mapping, as shown in Figure 2b), and modeled in a common space. The third one is to map the semantic embedding vector to the image feature space (called reverse mapping, as shown in Figure 2c) and identify unseen classes in the image feature space.

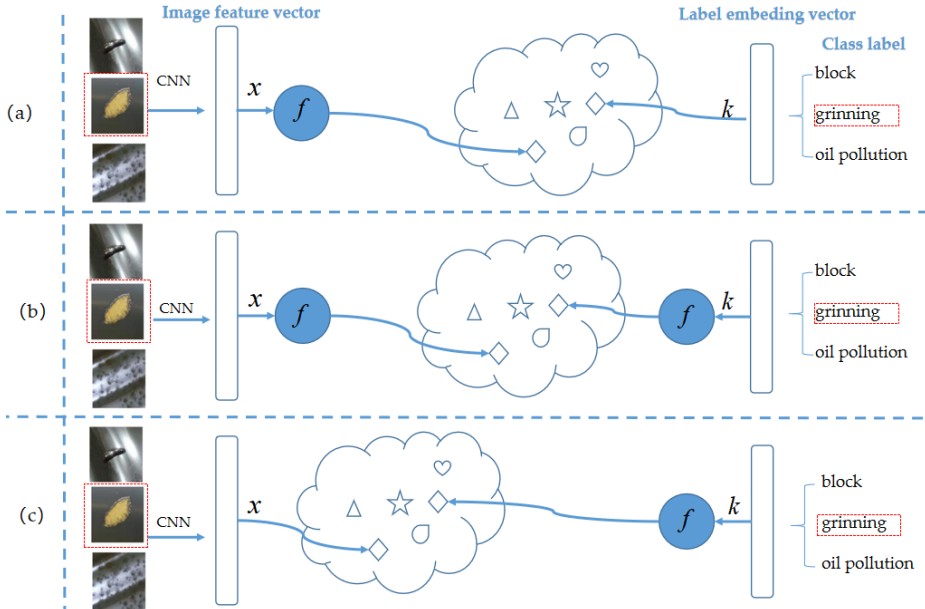

**Figure 2.** Visual semantic mapping methods. (**a**) Forward mapping, (**b**) Common mapping, (**c**) Reverse mapping

Among them, forward mapping has been a mainstream mapping method since the zero-shot learning problem was first applied in computer vision in 2009. In addition, common mapping first appeared in the form of a bilinear equation in 2013, and it has been widely adopted by researchers due to its good judgment and generalization ability. However, reverse mapping appeared in 2015 and has been on the rise. One of the reasons is that the image features extracted with deep learning technology have a good structure, and the same type of data are usually distributed in clusters, with stronger judgment ability. In contrast, the application time of reverse mapping in zero-shot learning is relatively short, and there is still a certain development space in the future.

Both forward mapping and common space mapping methods employ the embedding function of samples and semantic descriptors, which learn embedding by minimizing the similarity function between the sample and the corresponding semantic descriptor, with differences only in the embedding method and the selection of the similarity function. These methods, after embedding, are typically categorized using a nearest neighbor search.

However, in the high-dimensional space, the nearest neighbor search will suffer from the "hubness" problem (a certain number of data points will become the nearest neighbor or center of almost all test points), resulting in misclassification [36]. However, if the inverse mapping strategy [37] is adopted, namely mapping from the semantic space to the visual space, the hubness problem can be effectively avoided. Inspired by this article, we adopt the inverse mapping strategy in this paper. We further introduce the concept of relative features using pair relationships between data points. This not only provides additional structural information about the data but also reduces the dimension of the feature space, thereby reducing the hubness problem.

However, the reverse mapping strategy is learned from the data of seen classes, so it is inevitable to suffer from the problem of projection domain shift. Fu et al. [38] used multiple semantic information sources to explore the projection domain shift and implemented label transfer for unlabeled data of unseen classes to solve this problem. Recent research results,

Debasmit [39] developed a method of unsupervised domain adaptive, taking inspiration from which, correspondence between the projected semantic descriptors and unlabeled test data are explored, and then put forward a kind of unsupervised domain adaptive method based on local corresponding, this method is better than the global adaptation method.

We found that ZSL is usually an evaluation model only for unseen classes, however, in practical applications, seen classes often appear more frequently than unseen classes. Therefore, we expect the model to perform well for both seen and unseen classes, namely generalized zero-shot learning (GZSL) [40], therefore a calibration mechanism is developed to reduce the bias of seen class classification.

In this paper, aiming at the zero-shot recognition problem of wheel hub defect images, a new model is given through a three-step strategy to improve the problems: hubness, projection domain drift, and seen classes bias in generalized zero-shot recognition.

## 3. Wheel Hub Defect Dataset

### 3.1. Image Data

According to task requirements, we cooperated with local well-known hub manufacturers, collected hub defect image data on the production line, and completed defect type labeling according to the guidance of enterprise engineers and the defect standards of enterprises.

In order to eliminate the influence of the background and better extract the features of hub defect images, we automatically cut the target area, and then manually checked the segmented defect images to optimize the database. In this paper, nine kinds of hub defects (oil pollution, grinning, scratch, block, sagging, indentation, orange peel, deformation, and dust) were used to construct WHD-9, as shown in Figure 3.

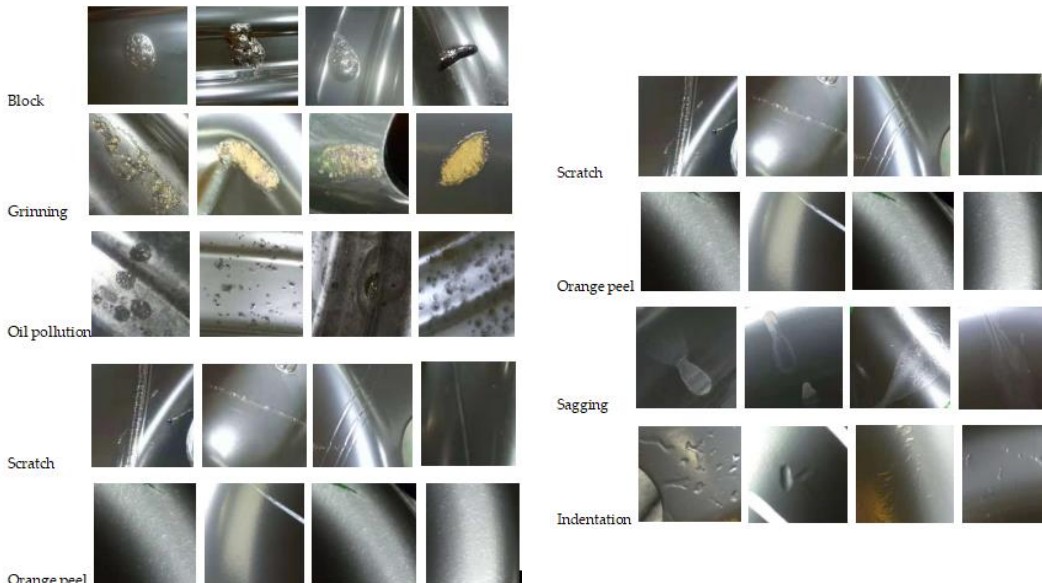

**Figure 3.** Images of wheel hub defects.

From Figure 4, we find that the distribution of classes is extremely uneven. The largest category is oil pollution defects, and the least number is orange peels. In this case, the machine learning algorithm finds it difficult to correctly identify the orange peel type. It is worth mentioning that some completely unknown defect types may appear in the future, which increases the difficulty of recognition, therefore, the study of this topic is very necessary.

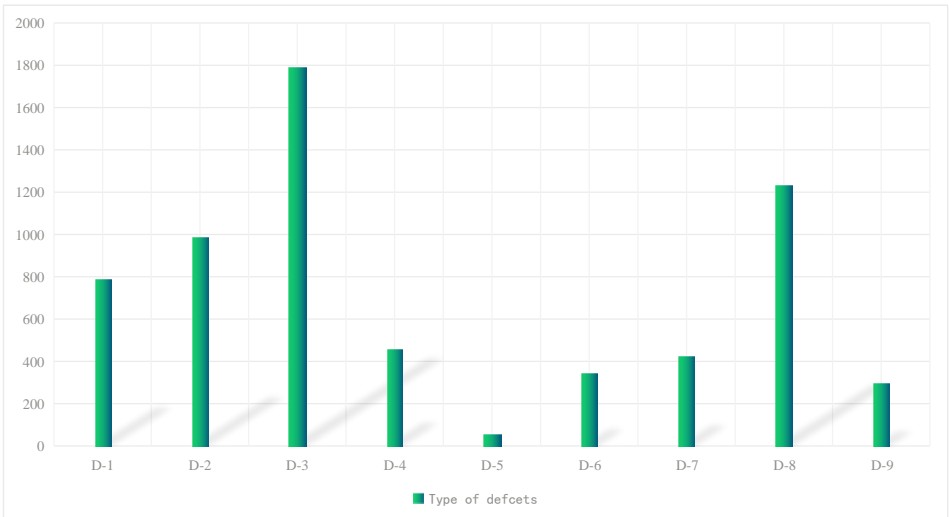

**Figure 4.** Sample distribution in WHD-9. Note: D-1 = block, D-2 = grinning, D-3 = oil pollution, D-4 = scratch, D-5 = orange peel, D-6 = sagging, D-7 = indentation, D-8 = dust, D-9 = deformation.

### 3.2. Domain Knowledge Base

The semantic attributes of each hub defect are different according to the product standards of the industry. As can be seen from Table 1, *block* is a lump on the surface of the paint film. *Dust* is white specks of soot that fall from the oven onto the surface of the wheel hub. *Oil pollution* is caused by mineral oil or grease attached to the metal surface, which is mostly round and dark in color with obvious protrusions, then 16 attributes were selected to describe the hub defect image in the semantic attribute space. Each defect type consists of a 16-dimension (A1, ..., A16) vector, and these vectors are encoded in one-hot [41] mode. These 16 dimension semantic vectors will be used to build a domain knowledge base for ZSL.

**Table 1.** Semantic attributes space for wheel hub defects.

| | Expert Defined Attribute | | | | | | | | | | | | | | | |
|---|---|---|---|---|---|---|---|---|---|---|---|---|---|---|---|---|
| | Color of Defects | | | Shape of Defects | | | | | | | Nature of Defects | | | | | |
| | A1 | A2 | A3 | A4 | A5 | A6 | A7 | A8 | A9 | A10 | A11 | A12 | A13 | A14 | A15 | A16 |
| D-1 | | | ✓ | | | | | ✓ | | ✓ | | | ✓ | | | |
| D-2 | | ✓ | | | | | | ✓ | | | | | ✓ | | ✓ | |
| D-3 | | | ✓ | | | ✓ | | | | | ✓ | ✓ | ✓ | | ✓ | |
| D-4 | ✓ | | | | | | ✓ | | | | | | ✓ | | ✓ | |
| D-5 | | | ✓ | | | | | ✓ | | | ✓ | ✓ | | | | ✓ |
| D-6 | | | ✓ | | ✓ | | | | | | ✓ | | ✓ | | | ✓ |
| D-7 | | | ✓ | | | | | ✓ | ✓ | | | | ✓ | | | ✓ |
| D-8 | ✓ | | | | | ✓ | | | | ✓ | | ✓ | | | ✓ | |
| D-9 | | | ✓ | | | | | ✓ | | | | | ✓ | ✓ | | |

Note: A1 = white, A2 = brown, A3 = dark gray, A4 = round, A5 = droplet shape, A6 = granular, A7 = linear, A8 = irregular shape, A9 = distinct sag, A10 = prominent protrusion A11 = obscure protrusion, A12 = dense distribution, A13 = sparse distribution, A14 = mostly distributed on the rim of steel ring, A15 = rough texture, 16 = smooth texture.

## 4. Method

### 4.1. Problem Description

The given training data set $D_{tr}$ is composed of $N_{tr}$ hub defect samples, so that $D_{tr} = \{(x_i, \quad a_i \quad y_i), \quad i = 1, 2, 3, \dots N_{tr}\}$. Here, $x_i \in R^{m \times n \times c}$ is the image sample ($m \times n$ is the image size, $c$ is the number of channels), $a_i \in R^s$ is the semantic descriptor of the defect category. Each semantic descriptor $a_i$ is associated with a unique defect label $y_i \in Y_{tr}$. The goal of ZSR is to predict the category label $y_j \in Y_{te}$ for the $j_{th}$ test sample defect $x_j$. In

a traditional ZSL, $Y_{tr} \cap Y_{te} = \varphi$, means that there is no overlap between seen and unseen defects. However, in the GZSL, test sets include not only unseen classes but also seen classes, that is $Y_{tr} \in Y_{te}$. During the training phase, semantic descriptors for both seen and unseen classes can be used. Since the probability of defects in seen classes is much higher than that of unseen classes in specific identification tasks, in this paper, a generalized ZSR framework was come up with to realize hub defect identification, as shown in Figure 5.

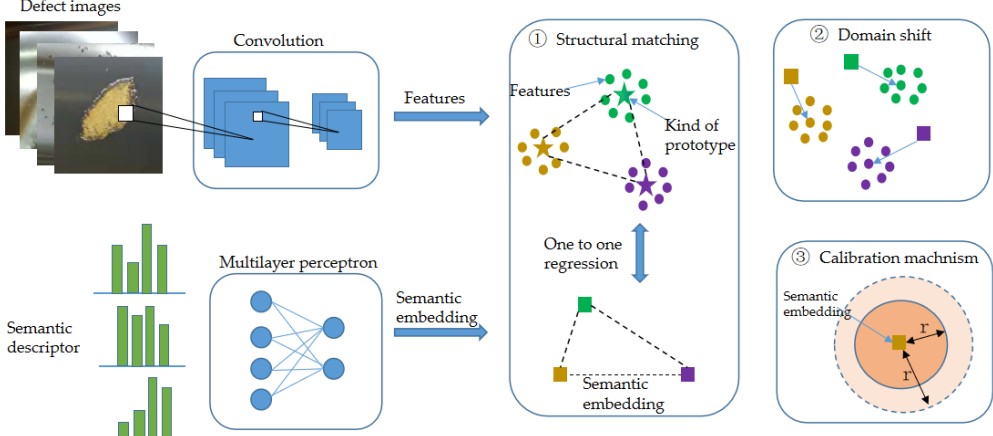

**Figure 5.** Generalized ZSR framework for wheel hub defect images.

As can be seen from Figure 5, a multilayer perceptron was adopted for the mapping of semantic descriptors to image feature space. Then the semantics were embedded into the corresponding features by means of one-to-one pairing. The semantics of the unseen class were then embedded to accommodate the test data of the unseen class. To avoid the framework's preference for seen class recognition, scaling calibration was performed during the test.

### 4.2. Structure Matching Strategy

About visual-semantic mapping, both forward mapping and common mapping methods use the embedding function of samples and semantic descriptors, and embedding processing is learned by minimizing the similarity function between the embedded sample and the corresponding embedded semantic descriptor. It is only different in the selection of embedding methods and similarity functions. These methods usually use nearest neighbor search for classification after embedding, however, in a high-dimensional space, nearest neighbor search always suffers from the "hubness" phenomenon, because a certain number of data points will become the nearest neighbors or centers of almost all test points, leading to classification errors. However, if the reverse mapping strategy is adopted, that is, the mapping from the semantic descriptor space to the visual feature space will effectively avoid the "hubness" problem, therefore, reverse mapping strategy was adopted in this paper.

We need to learn a mapping function $f(\cdot)$ the semantic descriptor $a_i$ to its corresponding image feature $\varphi(x_i)$. Where $x_i$ is the image, and $\varphi(\cdot)$ refers to the CNN architecture for extracting a high-dimensional feature map. The mapping function $f(\cdot)$ is a fully connected neural network. In order to make the descriptors and image features close to each other, the least squares loss function was employed to minimize the difference. The initial objective function $L_1$ is shown in Equation (1):

$$L_1 = \frac{1}{N_{tr}} \sum_{i=1}^{N_{tr}} \left\| f(a_i) - \varphi(x_i) \right\|_2^2 + \lambda_\gamma g(f) \tag{1}$$

where $g(\cdot)$ stands for normalized loss for $f(\cdot)$. The loss function $L_1$ was adopted to minimize the point-to-point difference between semantic descriptors and image features.

In order to illustrate the structural matching between the semantic space and the feature space, we tried to minimize the pairing relationships between classes in these two spaces. Therefore, we constructed relational matrices $D_a$ for semantic descriptors and image features, and each of these elements was derived from $[D_a]_{uv} = \|f(a^u) - f(a^v)\|_2^2$, where $a^u$ and $a^v$ represent semantic descriptors of seen class defects u and v respectively. The image feature relationship matrix $D_\varphi$ was built, and each element was calculated by the formula $[D_\varphi]_{uv} = \|\overset{-u}{\varphi} - \overset{-v}{\varphi}\|_2^2$. Where, $\overset{-u}{\varphi}$ and $\overset{-v}{\varphi}$ represent the mean values of class $u$ and $v$ respectively, which was calculated from Equation (2)

$$\overset{-u}{\varphi} = \frac{1}{|y_{tr}^u|} \sum_{y_i \in y_{tr}^u} \varphi(x_i) \tag{2}$$

where the $\sum(\cdot)$ is based on the defect type $u$, and $|y_{tr}^u|$ is the cardinality of the training set of defect type $u$, and the same is true for defect $v$.

To achieve structure alignment, the structure alignment loss function $L_2$ needs to be minimized.

$$L_2 = \|D_a - D_\varphi\|_F^2 \tag{3}$$

$\|\cdot\|_F^2$ Where represents the Frobenius norm, and combined with the loss functions $L_1$ and $L_2$, we get the total loss $L_{total}$, as Equation (4) shown.

$$L_{total} = L_1 + \rho L_2 \tag{4}$$

where $\rho \geq 0$, is used to measure the loss weight of $L_2$. $L_{total}$ is to optimize the parameters of $f(\cdot)$.

### 4.3. Domain Adaption Strategy

After training, projection domain shift may occur between the mapped semantic descriptors and the image features of the unseen classes. This is due to data from unseen classes not being used in the training phase, so regularized models have poor generalization ability to unseen classes. Therefore, we have to use test data from the unseen defect to adjust the mapping semantic descriptor to fit the unseen defects.

Given the mapping descriptors of unseen classes stack vertically in the form of a matrix $A \in R^{n_u \times d}$, where $n_u$ stands for the number of unseen classes, and $d$ means the dimension of semantic descriptor space.

Suppose $U \in R^{n_u \times d}$ is the test data set for the unseen class, and $O_u$ represents the number of test samples from the unseen class. To accommodate the mapped descriptors, a point-to-point correspondence between the descriptors and test data is used, which is represented as a matrix $C \in R^{n_u \times o_u}$. The rows of $U$ need to be rearranged so that each row of the revised matrix corresponds to the row in a$A$, which is achieved by minimizing the loss function Equation (5).

$$L_3 = \|CU - A\|_F^2 \tag{5}$$

$L_3$ can force $CU$ to generate adaptive semantic descriptors. However, there is still the problem that a sample may correspond to more than one descriptor in $A$, which means that it will actually result in corresponding to more than one category of test samples. To avoid this problem, by using group-Lasso [42], an additional group-based regularization function $L_4$ was conducted.

$$L_4 = \sum_j \sum_c \|[C]I_c j\|_2 \tag{6}$$

where $I_c$ means the index corresponds to those rows in the unseen class. So $[C]I_c j$ stands for a vector consisting of an index of rows $I_c$ and columns $j$. Since $C$ is a correspondence matrix, certain constraints can be applied to solve the deviation of sample number between

semantic space and feature space for unseen classes. Therefore, the domain adaptive optimization problem can be expressed as follows Equation (7):

$$\min_c \{L_3 + \lambda_g L_4\} \quad s.t. \quad C \geq 0, \quad C1_{O_u} = 1_{n_u}, \quad C^T 1_{n_u} = \frac{n_u}{o_u} 1_{o_u} \tag{7}$$

where $\lambda_g$ balances the weight of loss function $L_4$. The above optimization problems are all convex functions, which can be effectively settled by the conditional gradient method [43], and the method requires solving the linear function on the constraint $C \in D = \left\{ C : C \geq 0, \quad C1_{o_u} = 1_{n_u}, \quad C^T 1_{n_u} = \frac{n_u}{o_u} 1_{o_u} \right\}$ as an intermediate step, as shown in Algorithm 1. using the Simplex Formulation [44] in EMD [45], the variable $C_d$ in Algorithm 1 could be easily obtained [46].

---

**Algorithm 1:** Conditional Gradient Method

---

**Initialize:** $C_0 = \frac{1}{n_u o_u} 1_{n_u \times o_u}, \quad t = 1$
**Repeat**
$C_d = \mathrm{argmin} Tr(\nabla_{C=C_0}(L_3 + \lambda_g L_4)^T C), \ s.\,t.\ C \in D$
     $C_1 = C_0 + \alpha(C_d - C_0), \ for \ \alpha = \frac{2}{t+2}$
     $C_0 = C_1 \ and \ t = t + 1$
**Until** Convergence
**Output:** $C_0 = \mathrm{argmin}_C \{L_3 + \lambda_g L_4\} \ s.\,t.\ C \in D$

---

Algorithm 1 was used to obtain the final solution of the corresponding matrix $C_0$ and to check it. Given a test sample, we assigned the class correspondence to the maximum value of the corresponding variable and did the same for all test samples. The semantic descriptors of unseen classes were acquired by averaging features of the related classes, and then adaptive semantic descriptors were stacked vertically in the matrix $A'$.

*4.4. Recognition Anti-Bias Mechanism for GZSL*

In the GZSL [47], obviously, classification performance tends to favor seen defects. To eliminate this phenomenon, we recommend multiplication calibration for classification scores. In this paper, 1-nearest neighbor (1-NN) and Euclidean distance measure (EDM) were used as classifiers. For the test sample defect x, we adjusted the classification score of seen class as shown in Equation (8).

$$\hat{y} = \mathrm{argmin}\|x - f(a^c)\|_2 \cdot I[c \in \varphi] \tag{8}$$

where, if $c \in \varphi$ or $c \in U$, and $\varphi \cup U = T$, then $I[\cdot] = \gamma$. Here, $\varphi$, $U$, $T$ represent the seen defects, the unseen defects, and the collection of all defects, respectively. The scaling measure is to modify the effective variance of the seen defects. When the Euclidean distance metric is used for nearest neighbor classification, it assumes that the variance of all classes is equal, however unseen classes are not applied to learn the embedding space, changes in the characteristics of unseen classes are not considered, that is why the EDM is adjusted for seen classes only. For $\gamma > 1$, if we achieved a balance between the seen and unseen defects, which indicates that the variance of the seen classes was overestimated. Conversely, for $\gamma < 1$, a balance was found between seen and unseen classes, which means the variance of seen defects was underestimated. Algorithm 2 shows process of the proposed model from training to testing.

---

**Algorithm 2**: Three-step Zero-shot Learning Algorithm

---

**Input:** Training Dataset $\{x_i,\ a_i,\ y_i\}_{i=1}^{N_{tr}}$
**Parameters:** $\lambda_r,\ \rho,\ \lambda_g,\ \gamma$
**Repeat** (Training)
　　Sample Minibatch of $\{(x_i, a_i)\}$ pairs
　　Gradient descent $L_1 + \rho L_2$ w.r.t. parameters of $f(\cdot)$
**Until** Convergence (Step 1)
**Input:** Test Dataset $\{(x_i)\}_{i=1}^{N_{te}}$
　　Apply Algorithm 1 to obtain adapted descriptors of unseen classes $A'$ (Step 2)
**Repeat** for each test point x (Testing)
　　$\hat{y} = \underset{c \in T}{\arg\min} \|X - f(a^c)\|_2 \cdot I[c \in T]$ (Step 3)
**Until** all test points covered

---

## 5. Experiment and Analysis

### 5.1. Experiment Preparation

Based on the experimental setup, we evaluated using two data sets: aPY [48] including 20 seen classes and 12 unseen classes, and with an associated 64-dimensional semantic descriptor. WHD-9 is a self-built data set about wheel hub defects, cover 6 seen classes and 3 unseen classes, each associated with a 16-dimensional semantic descriptor. The details of both data sets are shown in Table 2.

**Table 2.** Dataset information.

|        | Training (Seen) | Testing (Unseen) | Attribute Dimension | No. of Samples |
|--------|-----------------|------------------|---------------------|----------------|
| aPY    | 20              | 10               | 65                  | 15,339         |
| WHD-9  | 6               | 3                | 16                  | 6380           |

With respect to the evaluation criteria, we used class-wise accuracy because it can avoid class dominance during intensive sampling. Therefore, the average precision of the class was calculated as follows Equation (9):

$$acc = \frac{1}{|y|} \sum_{y=1}^{|y|} \frac{the \quad number \quad of \quad correct \quad predictions \quad for \quad class \quad y}{the \quad total \quad number \quad of \quad samples \quad for \quad class \quad y} \tag{9}$$

where $|y|$ stands for the number of test defects. In the proposed model, the accuracy of seen and unseen classes is acquired respectively, and the harmonic mean $H$ is used for processing [49] as shown in Equation (10), which aims to ensure that the performance of the seen defect does not lead overall accuracy.

$$H = \frac{2 \times acc_s \times acc_u}{acc_s + acc_u} \tag{10}$$

where $acc_s$ and $acc_u$ are the classification accuracy of seen classes and unseen classes respectively. In order to make a fair comparison, we experimented and recorded the results for the training and test data sets on the common data sets and self-built data set.

### 5.2. Experiment Results

For the experiment, a two-layer feed-forward neural network for semantic embedding was employed. For the aPY and WHD-9 data sets, the dimensions of the hidden layer were selected as 1600 and 1200 respectively, and the activation function was ReLU. Image features were acquired by ResNet-101.

The proposed method was compared with previous ones. The first is to complete the baseline model (DEM) [35]. Then the DEM+R model only includes structural matching components using loss functions $L_2$ in the training phase. DEM+RA model includes structural loss components and domain adaptation components using loss functions $L3$ and $L4$. The DEM+ARC model includes all strategies: structural matching, domain adaptation, and anti-bias calibration. The parameters of the aPY, WHD-9 dataset ($\lambda_r$, $\rho$, $\lambda_g$, $\gamma$) were set to ($10^{-4}$, $10^{-1}$, $10^{-1}$, 1.1), and ($10^{-3}$, $10^{-1}$, $10^{-1}$, 1.1), respectively. In Table 3, class-wise accuracy results of the traditional unseen classes (TU), the generalized unseen classes (GU), the generalized seen classes (GS), and the generalized harmonic mean (H) are recorded.

**Table 3.** Experiment results based on aPY and WHD-9.

| | **aPY** | | | | **WHD-9** | | | |
|---|---|---|---|---|---|---|---|---|
| Methods | TU | GS | GU | H | TU | GS | GU | H |
| DAP[150] | 33.6 | 79.6 | 6.3 | 11.7 | 40.0 | 57.7 | 2.7 | 5.2 |
| IAP[140] | 36.9 | 67.3 | 7.5 | 13.5 | 31.7 | 65.1 | 1.9 | 3.7 |
| ConSE[141] | 28.7 | **92.7** | 0.0 | 0.0 | 37.2 | **83.2** | 2.1 | 4.1 |
| SYNC[142] | 24.3 | 68.1 | 9.1 | 16.1 | 45.6 | 67.8 | 9.3 | 16.4 |
| DeViSE[143] | **40.6** | 78.0 | 6.7 | 12.3 | 51.3 | 58.0 | 15.7 | 24.7 |
| DEM[102] | 36.4 | 76.9 | 13.2 | 22.5 | 48.7 | 63.4 | 20.4 | 30.9 |
| DEM+R(Our) | 31.2 | 72.3 | 17.1 | 27.6 | 50.0 | 59.7 | 23.4 | 33.6 |
| DEM+RA(Our) | 37.7 | 74.0 | 32.2 | 44.8 | 54.9 | 55.1 | 47.2 | 50.8 |
| DEM+RAC(Our) | 37.8 | 65.4 | **35.9** | **46.4** | **55.1** | 54.6 | **48.6** | **51.4** |

Note: $GS = acc_s$, $GU = acc_u$, H is obtained through Equation (10).

As shown in Table 3, the proposed method is more effective than previous popular methods, and compared with the baseline model, the harmonic mean of the proposed method was significantly increased. The performance improvement can be owed to the three-step strategy. For both data sets, only using structural matching (DEM+R) yielded better performance than the baseline model, with 22.7% (aPY) and 8.7% (WHD-9) improvements. Additional use of domain adaptation (DEM+RA) showed much better results than DEM+R, increased by 62.3% (aPY), 51.2% (WHD-9), but DEM+RAC with calibration components produced only marginal improvements, 3.5% (aPY), 1.2% (WHD-9). This is because the relational matrix-based component produces class-specific adaptation (DEM+RA) to the semantic embedding of unseen classes, whereas the calibration component is not category-specific, just distinguishing between seen classes and unseen classes, so it is understandable that the effect of improving performance is not obvious.

### 5.3. Experiment Analysis

(1)   Analysis of structure matching components

The effect of structure matching was analyzed by changing $\rho \in \{10^{-3}, 10^{-2}, 10^{-1}, 10^0, 10^1, 10^2\}$, and the change in accuracy was recorded. We conducted experiments using the WHD-9, the results of which are recorded in Figure 6.

It can be seen from Figure 6 that based on the WHD-9, the conventional unseen class accuracy (Figure 6a) and the generalized seen class accuracy (Figure 6b) are better than or similar to the baseline model DEM, while, the accuracy of the generalized unseen class (Figure 6c) is higher than that of DEM, and the harmonic mean (Figure 6d) is significantly better than that of DEM. It can be seen from Figure 6d that when $\rho = 10^1$, the effect of the harmonic mean is the best.

Compared with baseline DEM, we verified that the DEM+R strategy contributes to hubness reduction. Hubness is measured by a bias of 1 nearest neighbor histogram (N1) [50]. A smaller skewness of the N1 histogram means less hubness prediction, and we used test samples of unseen classes in the generalized setting, as shown in Table 4.

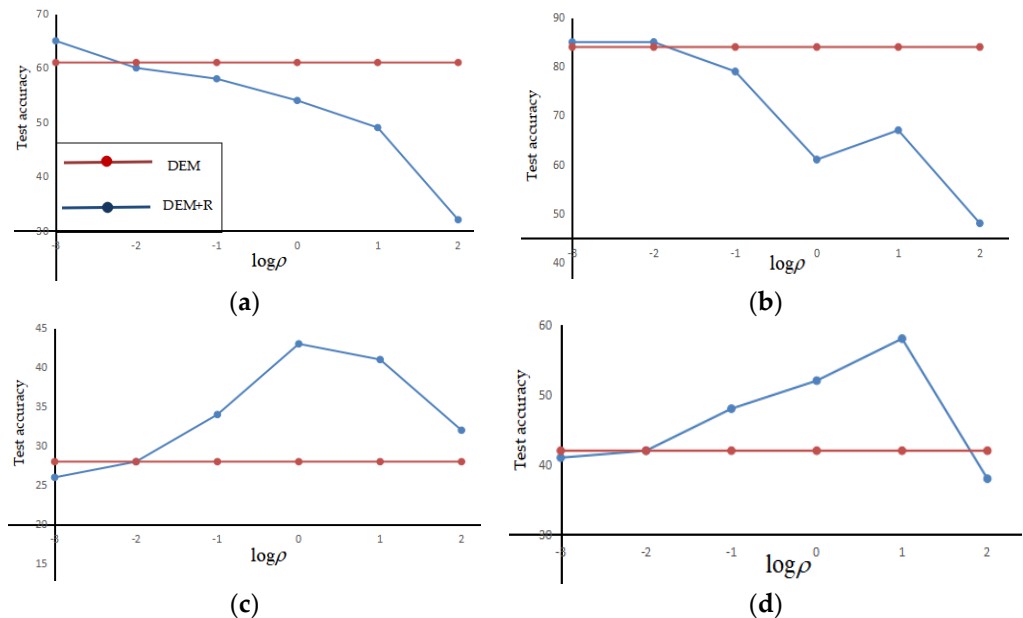

**Figure 6.** The accuracy change with $\rho$ on the WHD-9, baseline model (DEM) is represented in red, TU classes precision (**a**), GS classes precision (**b**), GU classes precision (**c**), and H precision (**d**) shows in blue.

**Table 4.** Hubness reduction using structure matching strategy.

| Method/Dataset | WHD-9 | aPY |
|---|---|---|
| DEM | 1.69 | 1.83 |
| DEM+R | 1.15 | 1.37 |

In Table 4, WHD-9 and aPY datasets are used for experiments of DEM and DEM+R. Let $\rho = 0.1$, the average value of multiple experiments is recorded in the Table, as can be seen from Table 4, the hubness of the N1 histogram generated by the DEM+R method on both data sets is smaller. This means that the use of an additional structural matching strategy will reduce hubness, therefore it can alleviate the trouble of dimension disaster.

(2)    Domain adaptive component analysis

As can be seen from Table 3, compared with DEM+R, the unseen class accuracy of DEM+RA increased by 88% (aPY), 101% (WHD-9), harmonic mean increased by 62.3% (aPY), 51.2% (WHD-9). Figure 7 shows the effect of domain adaptation by using t-SNE [51] on WHD-9. As shown in Figure 7a, the unseen class semantic embedding (purple) is very near to the seen class feature (blue). However, through the domain adaptation step, as shown in Figure 6b, the unseen class semantics are transformed, which is obviously close to the center of the feature family (red) of the unseen class.

(3)    Analysis of Calibration Mechanism

In Table 3, compared with the DEM+RA model, the "GU" performance of the DEM+RAC model was improved by 11.4%(aPY), 2.9% (WHD-9), and "H" performance increased by 3.6% (aPY), 1.2% (WHD-9), the reason may be that domain adaptive steps have already transferred the semantic embedding of unseen classes and accordingly shrunk the bias on seen classes and made further calibration not obvious.

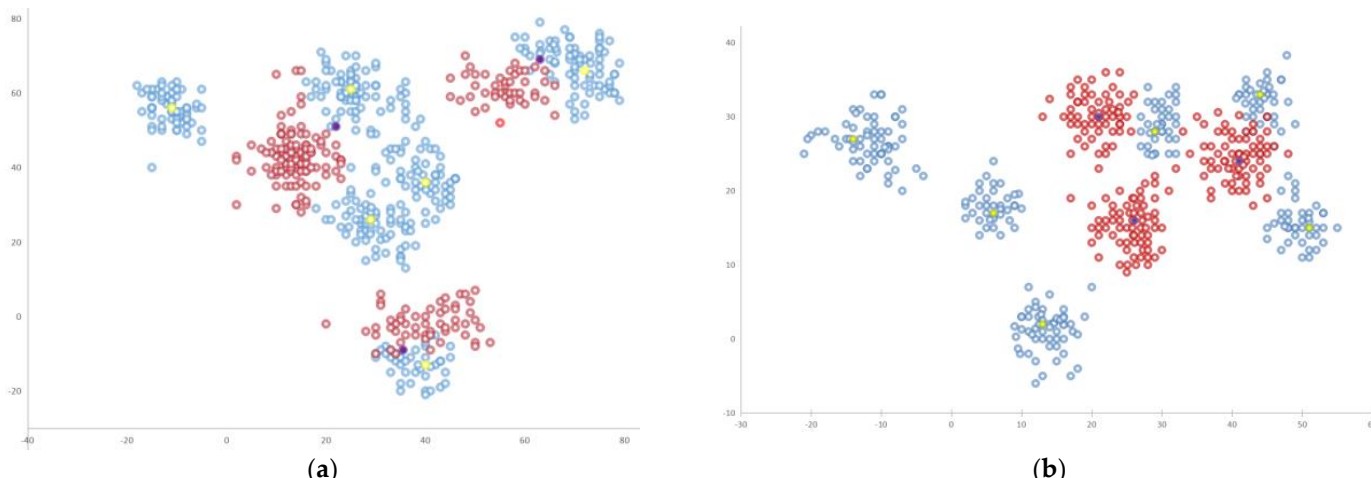

(**a**)　　　　　　　　　　　　　　　　　　　　　　(**b**)

**Figure 7.** t-SNE diagram of embedded samples. (**a**) Domain-free adaptation for WHD-9; (**b**) Domain adaptive operation is adopted. The image features of seen and unseen classes are represented in blue and red respectively. Embedded semantic descriptors for seen and unseen classes are represented in yellow and purple respectively.

### 6. Conclusions

In this paper, firstly a wheel defect data set (WHD-9) was built (image collection and domain knowledge expression). Secondly, a generalized zero-shot recognition framework for wheel hub defect image was proposed. This three-step recognition method is as follows: Step 1: Structural matching strategy, Step 2: Domain adaptation, Step 3: Calibration of classification scores. The model was validated by using the public data set aPY and the self-built WHD-9. The experiment result shows that the proposed three-step strategy is better than the previous method to a large extent, in that, Step 1 makes the hubness problem significantly reduced; Step 2 makes the projection domain shift well eliminated; Step 3 makes the bias problem of the seen class in the generalization recognition slightly decreased. Among them, the improvement of Step 2 is the most obvious.

**Author Contributions:** Conceptualization, X.S.; Data curation, X.S., Y.M.; Formal analysis, X.S., H.S.; Funding acquisition, J.G., M.W.; Investigation, X.S., Y.M.; Methodology, X.S.; Resources, X.S., H.S.; Software, X.S., H.S.; Supervision, J.G., M.W.; Writing—original draft, X.S.; Writing—review & editing, X.S., J.G., M.W. All authors have read and agreed to the published version of the manuscript.

**Funding:** This research was funded by the National Natural Science Foundation of China (No. 51875266), Key projects of Science and Technology of Henan Province (No. 202102110114).

**Institutional Review Board Statement:** Not applicable.

**Informed Consent Statement:** Not applicable.

**Data Availability Statement:** The data presented in this study are available on request from the corresponding author. The data are not publicly available due to the company where the defect image came from hopes me keep a secret.

**Conflicts of Interest:** The authors declare no conflict of interest.

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
