# Peer review of "Wheel Hub Defects Image Recognition Base on Zero-Shot Learning"

_applsci, doi:10.3390/app11041529_

Round 1

Reviewer 1 Report

  1. Grammar mistakes and typos, for example:
    • In abstract, line 6: Therefore, in this paper, a generalized zero-shot learning framework for hub defect image recognition.
    • Page 2, 2nd paragraph, line 2: In order to identify unseen classes, we usually use a large labeled samples from
    • page 16, line 2: And in Figure 7 shows the effect of domain adaptation by using
    • page 16, 2nd paragraph: It can be seen from Table 3 that the performance improvement between DEM+RAC and DEM+RA is GU by 11.4
  2. The problem "Wheel Hub Defects recognition" is not explained in the introduction.
  3. The literature review is also very generic, it lacks the previous work done on the specific problem of "wheel hub defects recognition".
  4. What is the originality of this work?
  5. The author's contribution is not clear.
  6. In table 3. harmonic mean H is used to report the performance. It is not explained whether the accuracy used for computing H includes both seen and unseen classes or just unseen classes.

Reviewer 2 Report

The major problem of this paper is writing. The motivation is not clear as well as the contribution. More details are:

1. If semantic descriptor exists, would it possible to use the semantic descriptor to determine the defect types ?

2. If you has some information of unseen defects and extract features from that information, will the be treated as zero shot learning ?

3. Why we need reverse mapping?

4. Writing format has problem at Section 4.2. E.g. corressp_o_nding

5. In alg2 box, it is three-step algorithm. However, you can not find any three steps in the bounding box. 

6. Reference 6 is missing key information. 

Round 2

Reviewer 1 Report

I appreciate the changes authors have made in response to the raised concerns. 
I am satisfied with author's response and find it appropriate to be accepted to this journal.